# Predicting the remaining useful life of metro pantograph sliding strips using gamma processes and its implications for maintenance scheduling

Jie Liu[1]*, Chuang Wu[2]

1 School of Intelligent Manufacturing and Transportation, Chongqing Vocational Institute of Engineering, Chongqing, China, 2 Maintenance Department, Chongqing Rail Transit (Group) Co.Ltd, Chongqing, China

☯ These authors contributed equally to this work.
* 943069788@qq.com

## Abstract

The progressive wear of pantograph sliding strips on metro trains necessitates timely replacement to ensure safe and reliable operations. This study proposes an adaptive, data-driven framework for predicting the remaining useful life (RUL) of these components, leveraging operational data from Chongqing Metro Line 6. A Gamma-process model is employed to capture the wear behavior under real-world operating conditions, integrating historical records and new observations through Bayesian inference. Markov chain Monte Carlo (MCMC) sampling is then applied to solve the posterior distribution, with three parameter-estimation approaches compared and the model's predictive accuracy evaluated across different life-cycle stages. The results demonstrate that incorporating prior knowledge significantly improves prediction accuracy. To showcase practical utility, the study devises a maintenance-scheduling strategy that integrates RUL forecasts with regular vehicle-maintenance intervals, thereby extending service life and reducing costs. Validated using real-world data, the proposed methodology offers a pragmatic tool for predictive maintenance in metro systems and can be adapted to similar engineering applications.

## 1 Introduction

Sliding strips are critical components in Metro pantograph systems, enabling the transfer of electrical power from overhead contact lines to trains. The widely employed Steemann single-arm pantograph (Fig 1) incorporates two interchangeable carbon sliding strips in its upper structure. These strips experience ongoing deterioration due to frictional wear, electrical arcing, and mechanical stress, necessitating replacement when their thickness diminishes to approximately 24.5 mm. Fig 2 illustrates the degradation process, contrasting the initial and worn profiles of

**Data availability statement:** All data and code are publicly available on GitHub (https://github.com/jery761210/sliding-strips.git);

**Funding:** This work was supported by the Science Technology Research Program of Chongqing Municipal Education Commission (Grant No. KJQN202203405, KJQN202303402, KJQN202203417). The funders had no role in study design, data collection and analysis, decision to publish, or preparation of the manuscript.

**Competing interests:** The authors have declared that no competing interests exist.

two decommissioned strips, underscoring the substantial material erosion observed during operation.

The current maintenance protocols face substantial challenges. Technicians frequently conduct manual thickness assessments during inspections, relying on subjective judgment to determine the need for replacing strips approaching the 24.5 mm threshold. This approach often leads to premature replacements and suboptimal strip utilization. The dataset presented in Fig 3 reveals that out of 1,120 wear-related replacements, 368 strips (32.9%) continued functioning below the threshold, 30 strips (2.7%) were replaced precisely at the threshold, and 722 strips (64.4%) were replaced while still above it. This distribution highlights significant inefficiencies, with approximately 65% of strips replaced prematurely and about 33% operating beyond the recommended limit.

The present study sought to develop predictive models for forecasting the degradation of pantograph sliding strips and estimating their RUL, defined as the time until a strip reaches a critical wear threshold. Accurate RUL estimation can facilitate maintenance planning that aligns with the actual wear progression, thereby reducing waste and enhancing economic efficiency.

Existing research on pantograph strip wear has primarily focused on predicting strip thickness or classifying its condition, rather than directly estimating the RUL.

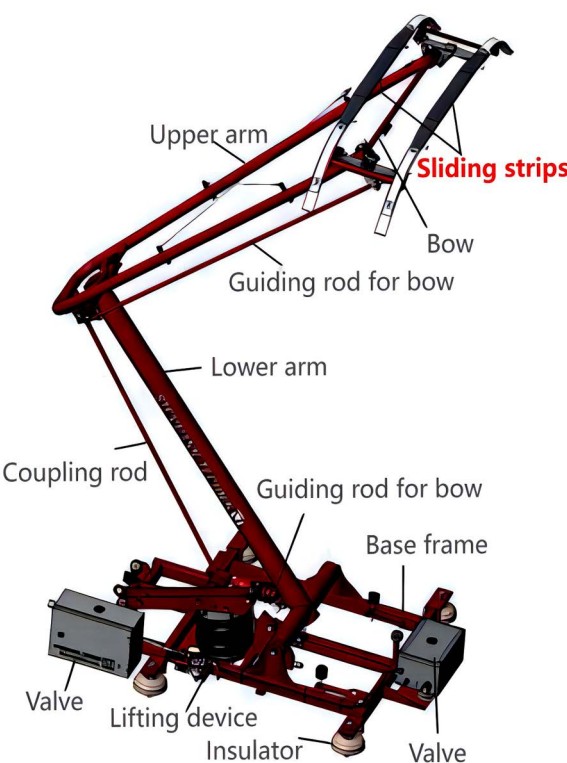

**Fig 1. Configuration of the Steemann single-arm pantograph on Chongqing Metro Line 6.** The highlighted area shows the designated position for the sliding strip installation.

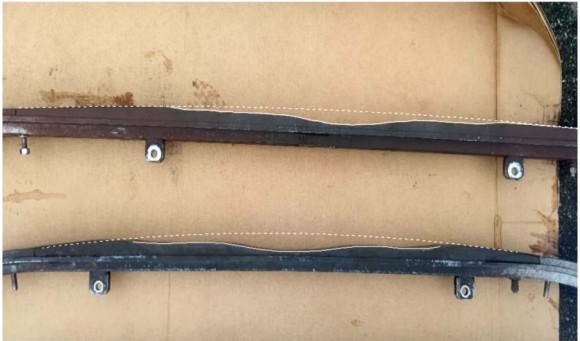

**Fig 2. Worn patterns on retired sliding strips.** The dashed line represents the original profile, and the solid line indicates the current profile.This study analyzed thickness data from 1,207 sliding strips extracted from 20 trains operating on Chongqing Metro Line 6 between 2018 and 2021. The dataset comprehensively documented the thickness evolution of each strip from initial installation to eventual removal. The findings revealed that only 7% of replacements were due to critical issues like melting or cracking, while the remaining 93% were attributed to regular wear and tear. Consequently, the analysis focused on wear-related replacements, which constituted the vast majority of maintenance interventions.

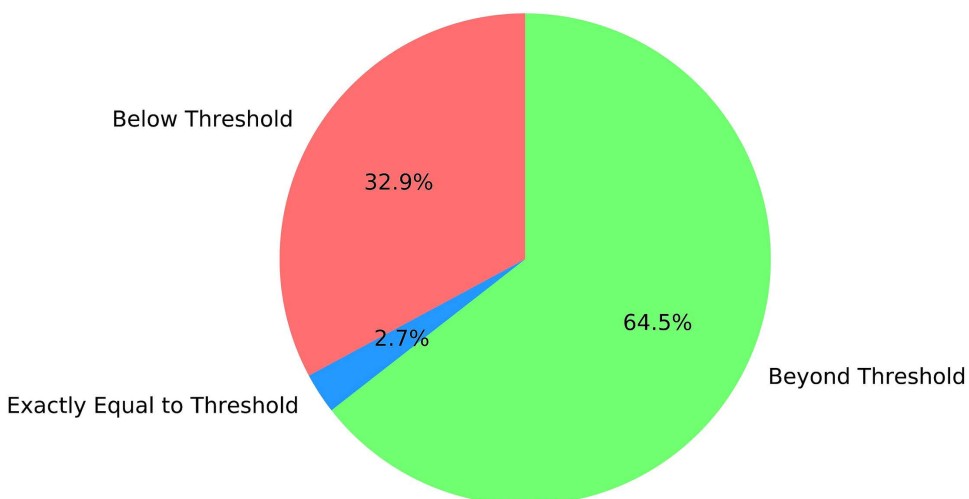

**Fig 3. Distribution of strip thickness at replacement.** The red sector represents strips below the threshold, blue indicates strips at the threshold, and green represents strips above the threshold.The economic implications of the current pantograph sliding strip replacement practices are substantial. The average service life of these components is only 71 days, with each replacement costing approximately 5,000 RMB, resulting in significant maintenance expenditures. Field data indicate that 64.5% of the replaced strips still possessed adequate thickness for continued safe operation, suggesting potential cost savings through optimized replacement timing. Conversely, 33% of the strips exceeded the recommended threshold between 2018 and 2021 without reported safety incidents, though the safety implications of extended use were not assessed in this study.

Kuźnar et al. [1] utilized artificial neural networks to forecast strip thickness with less than 12% error. Xu et al. [2] achieved 97.3% accuracy in wear trend fitting using support vector regression, while Trilla et al. [3] reduced thickness prediction uncertainty to 1.39 mm by employing a time delay neural network ensemble that integrates seasonal variables. Classification methodologies have also been explored, with Kuźnar et al. [4] and Kuźnar and Lorenc [5] demonstrating high accuracy in categorizing strip conditions.

The existing literature exhibits a dearth of studies aimed at predicting the RUL of pantograph sliding strips. To date, only two notable contributions have been identified in this domain. Guan et al. [6] proposed a Wiener process model

tailored to address the challenge posed by unevenly spaced wear data in carbon contact strips of railway pantographs. Their methodology integrated the Wiener process with a weighted linear regression model for unevenly spaced intervals, effectively predicting RUL using authentic operational data from the Guangzhou Metro. Conversely, Li et al. [7] developed a reliability-centered framework for predicting RUL by investigating wear patterns under varying contact pressure fluctuations. Through experiments on sliding electrical contacts, they formulated both a longevity prediction model and a probability distribution model for the phase of stable wear.

Despite their potential, both methods exhibit notable limitations that hinder their practical application. The statistical assumptions underlying these models often fail to adequately capture the complexities of real-world scenarios, and they do not seamlessly integrate with current maintenance planning systems. This disconnect between theoretical frameworks and practical needs substantially diminishes their utility for informing maintenance decisions. Moreover, the accuracy of RUL predictions is highly sensitive to the precision of thickness measurements, posing a significant challenge. A mere 1 mm discrepancy in thickness prediction can result in a 33-day deviation in RUL estimation based on typical wear rates. Such considerable inaccuracies could lead to premature replacements or, even worse, unforeseen failures that jeopardize service reliability.

This study addresses the aforementioned challenges through two key contributions. Firstly, we introduce a data-driven prognostic methodology that employs a gamma stochastic process to characterize the progressive wear patterns of sliding strips and forecast their RUL. This approach effectively captures the inherent variability in wear and produces reliable RUL predictions through Monte Carlo sampling, offering a viable alternative to conventional Wiener process models [6]. Secondly, we propose a maintenance optimization strategy that synchronizes strip replacements with routine train maintenance schedules, optimizing strip utilization while upholding safety standards. We provide a quantitative evaluation of the economic advantages derived from this optimized scheduling strategy. These contributions collectively establish a comprehensive RUL-centered maintenance framework that advances methodological development and enhances practical utility in transit operations, significantly improving maintenance efficacy and decision-making precision.

The paper is structured as follows. Section 2 analyzes the dataset characteristics and justifies the selected degradation model. Section 3 Construct experimental data and preprocess all data.Section 4 introduces the gamma process for predicting RUL and validates the prediction accuracy. Section 5 demonstrates how RUL predictions can enhance replacement scheduling and quantifies the associated cost savings. Section 6 investigates the variation in prediction accuracy with and without prior data, providing practical implementation insights and an in-depth discussion. Finally, Section 7 concludes the study and proposes potential future research directions.

## 2 Stochastic degradation model selection and validation

### 2.1 Review of remaining useful life estimation methodologies

Accurately predicting the RUL of a sliding strip before it reaches a critical wear threshold is essential for efficient predictive maintenance. Current RUL estimation techniques fall into three primary categories: model-based approaches [8], data-driven methods [9], and hybrid methodologies [10]. Model-based strategies utilize analytical or physical models to predict component behavior and degradation, providing precise estimates without requiring extensive empirical data. Nonetheless, accurately modeling the intricate stochastic degradation and interactions among various factors in a purely physics-based model poses notable challenges.

The increased focus on data collection in operational processes has resulted in a rise in the popularity of data-driven techniques. These approaches utilize historical monitoring data along with statistical or machine learning algorithms to model degradation patterns, reducing the need for detailed physical understanding. While effective, data-driven methods typically necessitate extensive datasets, and their predictive accuracy may decrease when real-world operating conditions differ significantly from the historical training data. Hybrid methodologies, which combine aspects of both data-driven

and physics-based approaches, seek to achieve a balance between accuracy and data quantity. Nevertheless, existing research has not definitively proven that hybrid methods consistently outperform either individual approach.

The effectiveness of pantograph sliding strips is impacted by various factors such as sliding speed, current collection intensity, relative height to the contact wire, and contact force [11]. The complex interaction among these parameters presents a substantial obstacle in formulating a precise theoretical degradation model. Nevertheless, the wealth of empirical data obtained from regular pantograph strip assessments establishes a solid basis for data-driven methodologies.

Estimation of RUL has been approached using diverse data-driven methodologies, such as regression models, similarity-based techniques, stochastic process models, and deep learning methods [12–21]. Researchers have introduced random coefficient regression frameworks to characterize degradation patterns and deduce lifetime distributions from data obtained through condition monitoring [12]. In cases where failure records are lacking, analysts may redefine the prognostic task or apply a Bernstein distribution to establish priors for random coefficients [13,14]. Similarity-based approaches involve identifying reference trajectories that closely resemble the unit under examination and calculating RUL as a weighted average of the most similar historical instances [15–18]. However, these methods often depend on manually defined similarity thresholds that may lack generalizability across different operational settings. On the other hand, stochastic process models, such as Wiener process models, have shown superior performance in capturing the inherent variability in degradation [19]. Various deep learning architectures, including autoencoders, deep belief networks, convolutional neural networks, and recurrent neural networks, have been utilized in equipment health monitoring for prognostic purposes [20]. Despite their robust nonlinear modeling capabilities, deep learning techniques necessitate substantial computational resources and extensive labeled datasets, which could impede their practical application. The existing literature does not provide a definitive conclusion on whether deep learning models consistently outperform traditional prognostic methods in predicting RUL, particularly in scenarios with limited failure data [21].

Researchers have utilized Wiener [22,23], Gamma [24,25], and Inverse Gaussian (IG) [26,27] processes in stochastic process models to characterize the deterioration of engineering components. The Gamma and IG processes, classified as Lévy processes with parameters, generate degradation paths with consistently positive increments, making them well-suited for modeling processes like wear or crack growth involving irreversible damage accumulation [24]. In contrast to the conventional stochastic model driven by Brownian motion, which can capture non-monotonic degradation and provide an approximate representation of monotonic degradation, offering considerable flexibility. For instance, a nonlinear Wiener process has been applied to predict bearing conditions, effectively capturing non-monotonic trends in vibration-based health indicators [28]. Hence, our research employs a stochastic process model based on a comprehensive review of existing literature. While all three processes can simulate wear progression, selecting the most suitable model hinges on the data characteristics, particularly the presence of monotonic patterns in the degradation trajectory and the optimal probability distribution for modeling wear increments. Although both Gamma and IG processes ensure monotonic paths, they diverge in their assumptions regarding increment distributions (Gamma distribution for Gamma processes and IG distribution for IG processes). The choice between these models should be guided by empirical validation to ascertain which increment distribution best aligns with the observed data.

## 2.2 Empirical analysis of wear monotonicity characteristics

The pantograph sliding strips inevitably experience thinning over time due to irreversible material loss from wear. Our wear analysis corroborates this trend, with data indicating either a steady decrease or constancy in strip thickness. As illustrated in Fig 4, the wear trajectories for twenty strips consistently demonstrate a decline in thickness, although minor deviations may arise from measurement errors. This consistent wear pattern represents a critical aspect that must be incorporated into any robust degradation model [29].

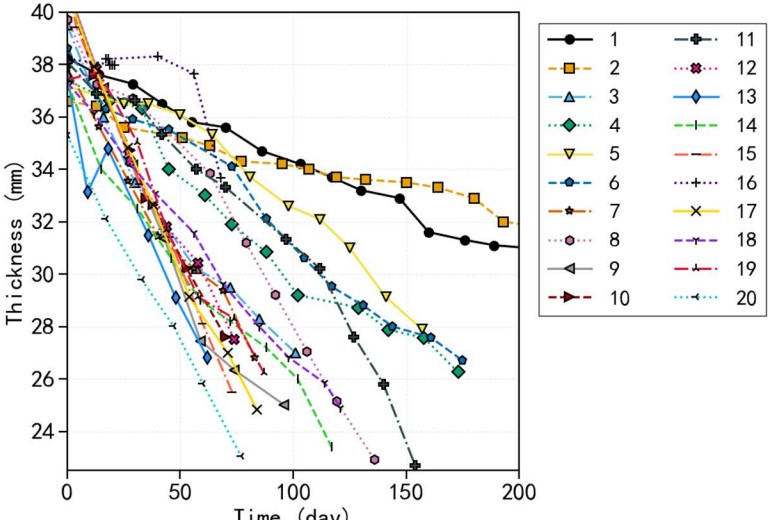

**Fig 4. Wear trajectories of individual pantograph sliding strips over time.** The figure shows twenty typical wear trajectories, demonstrating a consistent decrease in thickness.

## 2.3 Goodness-of-fit test for wear increments

The task at hand involves the selection of an appropriate probability distribution for these increments. Two potential distributions, namely Gamma and Inverse Gaussian (IG), were evaluated using Anderson–Darling (AD) goodness of fit tests [30]. Each strip's positive wear increments were analyzed by fitting both a Gamma distribution and an IG distribution. Subsequently, the AD test was employed to compare the observed increment data with the cumulative distribution function (CDF) of each fitted distribution. The null hypothesis posited that the increments in the strip follow the specified distribution. If, at a significance level of 5%, the AD test indicated no statistically significant deviation between the empirical data and the fitted CDF, the distribution was considered a suitable model for the strip's increments, demonstrating a successful outcome of the goodness of fit test for that distribution. Conversely, if the AD test detected a significant deviation at the 5% level, the distribution was rejected as an inadequate fit for that strip.

Table 1 displays the goodness-of-fit outcomes for wear increment data under two distributional assumptions. The Gamma distribution notably fits the wear increment data well across the entire fleet. Individual train pass rates for the Gamma distribution range from approximately 89% to 100%, with an overall 97.9% of the 1,120 increment series showing no significant deviation from the fitted Gamma model (Column 3 in Table 1). Specifically, only 18 series were deemed incongruent with the Gamma distribution by the Anderson-Darling (AD) test. Almost all trains adhered to the Gamma distribution, with 17 out of 20 trains having at least 95% of their strips passing the Gamma test; the one exception still achieved around 89%. This strong support highlights the excellent fit of wear increments with the Gamma distribution in nearly all cases.

While the Inverse Gaussian distribution achieved an impressive overall pass rate of 95.8%, its consistency across the entire fleet was somewhat lower. Individual train pass rates varied from 83% to 100% under this model, with 38 series showing significant deviations, approximately double the outliers compared to the Gamma distribution. It is notable that some trains, which had perfect pass rates under the Gamma distribution, encountered failures in one or two instances when tested using the Inverse Gaussian model. This highlights the superior ability of the Gamma distribution to capture the full range of observed wear increment patterns.

**Table 1. Goodness-of-fit of gamma and inverse Gaussian wear increments across train datasets.**

| Train Number | n | Gamma distribution pass (%) | Inverse Gaussian distribution pass (%) |
|---|---|---|---|
| 1 | 44 | 90.9 | 90.9 |
| 2 | 46 | 97.8 | 95.7 |
| 3 | 44 | 100 | 95.5 |
| 4 | 37 | 89.2 | 83.8 |
| 5 | 31 | 90.3 | 83.9 |
| 6 | 40 | 95 | 82.5 |
| 7 | 34 | 100 | 97.1 |
| 8 | 40 | 100 | 100 |
| 9 | 52 | 100 | 100 |
| 10 | 78 | 96.2 | 96.2 |
| 11 | 79 | 100 | 100 |
| 12 | 76 | 100 | 94.7 |
| 13 | 42 | 100 | 100 |
| 14 | 75 | 100 | 100 |
| 15 | 76 | 98.7 | 97.4 |
| 16 | 72 | 100 | 100 |
| 17 | 79 | 100 | 100 |
| 18 | 73 | 100 | 100 |
| 19 | 65 | 100 | 98.5 |
| 20 | 37 | 100 | 100 |
| Overall | 112020 | 97.9 | 95.8 |

After comprehensive analysis, the Gamma process emerges as the optimal stochastic model for sliding-strip wear. Among the three processes evaluated (Wiener, Gamma, and IG), only the Gamma and IG processes exhibit consistent sign increments, leading to monotonically decreasing paths. Although the Wiener process is mathematically tractable, it may yield unrealistic thickness fluctuations when unrestricted. Moreover, the wear increment distribution associated with the Gamma distribution, boasting a 97.9% success rate, aligns both theoretically and empirically with models of sliding strip degradation.

## 3 Experimental dataset construction and preprocessing methodology

### 3.1 Analysis of factors influencing wear rate and representative sample selection

A test dataset was created using wear records to illustrate the methodology for predicting Remaining Useful Life (RUL) and planning maintenance. The dataset was structured based on train number, strip position, and manufacturer. Information from all 20 trains operating on Chongqing Metro Line 6 was gathered for this investigation, covering the complete fleet. As depicted in Fig 5, every six-car train in the fleet is fitted with two pantographs, each containing two sliding strips, totaling four strips per train (designated as positions 1–4). These strips were supplied by four distinct manufacturers (designated as 1–4).

To preserve fleet diversity and facilitate analysis, we selected a single strip randomly from each train, yielding 20 representative samples. Table 2 presents information on the train, position, and manufacturer of each sample, while Fig 8 depicts the thickness-time trajectories corresponding to these samples, all demonstrating a uniform, gradual reduction in thickness.

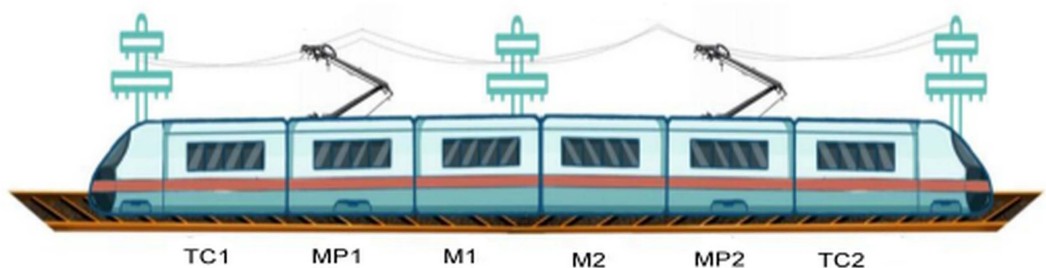

**Fig 5. Train configuration with six cars.** The trailer cars ("TC") are at both ends, and the four intermediate cars serve as motor cars. Two motor cars ("MP1" and "MP2") are equipped with roof-mounted pantographs, while the other two ("M1" and "M2") lack pantographs.To obtain a representative sample, we first identified variables influencing the rate of sliding-strip wear, defined as the average daily thickness reduction per strip. A one-way ANOVA analysis, followed by a Compact Letter Display post-hoc assessment, unveiled significant differences among the 20 trains [31,32]. Fig 6 classifies the fleet into three distinct groups based on wear rates, demonstrating statistical significance. These variances are likely linked to operational factors such as the duty cycle and average operating speed of each train.

**Table 2. Characteristics of 20 train sliding strips: number, position, manufacturer.**

| Train Number | Position | Manufacturer | Train Number | Position | Manufacturer |
|---|---|---|---|---|---|
| 1 | 1 | 3 | 11 | 2 | 4 |
| 2 | 2 | 3 | 12 | 3 | 4 |
| 3 | 4 | 3 | 13 | 1 | 3 |
| 4 | 1 | 3 | 14 | 3 | 4 |
| 5 | 1 | 2 | 15 | 2 | 4 |
| 6 | 3 | 3 | 16 | 2 | 4 |
| 7 | 4 | 1 | 17 | 1 | 4 |
| 8 | 4 | 4 | 18 | 4 | 3 |
| 9 | 3 | 3 | 19 | 3 | 4 |
| 10 | 4 | 4 | 20 | 3 | 4 |

### 3.2 Degradation data transformation

In practical scenarios, raw thickness measurements commonly show a declining pattern over time, in contrast to the anticipated increasing degradation measure in a conventional Gamma process model [22]. To reconcile this disparity, we propose the incorporation of a linear transformation to the thickness data of individual strips. Let $x_{i1}$ denote the maximum recorded thickness of strip i, and $\varepsilon$ represent the wear threshold. The transformed degradation indicator $y_{ij}$ for the j-th observation of strip i is formulated as.

$$y_{ij} = 1 - \frac{x_{ij} - \varepsilon}{x_{i1} - \varepsilon} \quad i = 1, \cdots\cdots, m, \, j = 1, \cdots\cdots, n_i$$

(1)

Let $m$ represent the total number of strips in the sample, and $n_i$ denote the number of observations for strip i. This conversion standardizes the data of each strip to a scale ranging from to 1: at installation $y_{ij} = 0$, and upon reaching the threshold $y_{ij} = 1$ (or $y_{ij} > 1$ if the threshold is surpassed).

Fig 9 depicts normalized wear accumulation paths, starting at zero and reaching one gradually; values exceeding one signify excessive wear. A consistent preprocessing methodology is employed for all thickness data to uphold data integrity and computational dependability.

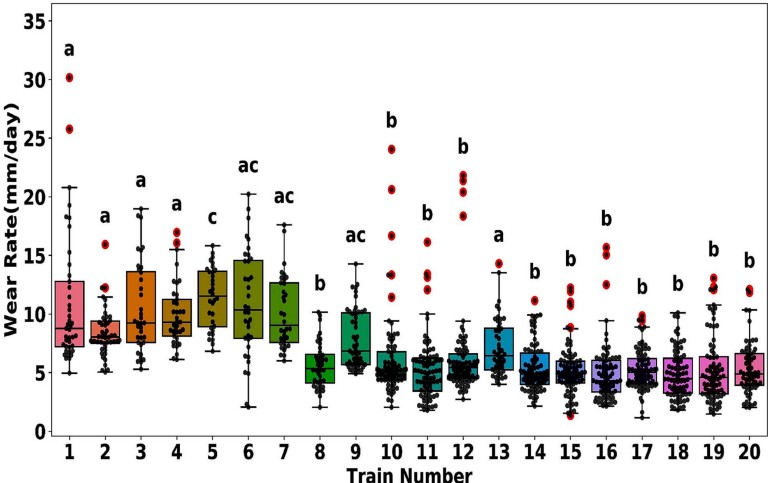

**Fig 6. Box-and-whisker plots showing the mean daily wear rate of 20 trains.** The x-axis represents train numbers, and the y-axis shows the wear rate in millimeters per day. Outliers are marked with red dots. Compact Letter Display labels above each box indicate statistical groupings: identical letters denote nonsignificant differences, while distinct letters represent significant differences among trains. Fig 7 illustrates consistent wear rates among strips 1–4 within a single train, suggesting uniform rates of wear within the same train.

## 4 Gamma process modeling and bayesian estimation of remaining useful life

### 4.1 Gamma process model selection via hypothesis testing

This integrated methodology merges historical degradation patterns identified in Section 3 with present monitoring data within a Bayesian framework. This enables the ongoing adjustment of model parameters to uphold the dependability of predictive maintenance. The algorithm chooses historical trajectories by aligning static attributes (e.g., train identity and strip position) with those of each pending strip. As depicted in Fig 10, each pending strip aligns with three to seventeen corresponding historical replacements.

The probability density function (PDF) governing the wear increment $\Delta y$ in a conventional Gamma degradation model is specified by Equation (2) [29]

$$f(\Delta y) = \frac{\beta^{\alpha\Delta t}\Delta y^{\alpha\Delta t-1}e^{-\beta\Delta y}}{\Gamma(\alpha\Delta t)}$$

(2)

The model utilizes shape and scale parameters, denoted as α and β, respectively, with $\Gamma(\textbf{.})$ the Gamma function represented by $\Delta y$. In contrast, the random-effect Gamma process model incorporates variability in the degradation rate among items by treating it as a random variable. Consequently, the integrated comprehensive PDF of undefined accounts for this random effect [35]

$$f(\Delta y) = \int_0^{+\infty} f(\Delta y)g(\beta;\eta,\delta)d\beta = \frac{\Delta y^{\alpha\Delta t-1}\delta^\eta\Gamma(\alpha\Delta t+\eta)}{\Gamma(\alpha\Delta t)\Gamma(\eta)(\Delta y+\delta)^{\alpha\Delta t+\eta}}$$

(3)

The PDF representing a Gamma distribution with shape parameter η and scale parameter δ is denoted as $g(\beta;\eta,\delta)$. This extended model accounts for the variability in wear rates at the individual level. Although more complex versions of the Gamma process have been suggested [36], their heightened intricacy could hinder parameter estimation. Therefore, we choose the aforementioned simpler formulation for this study.

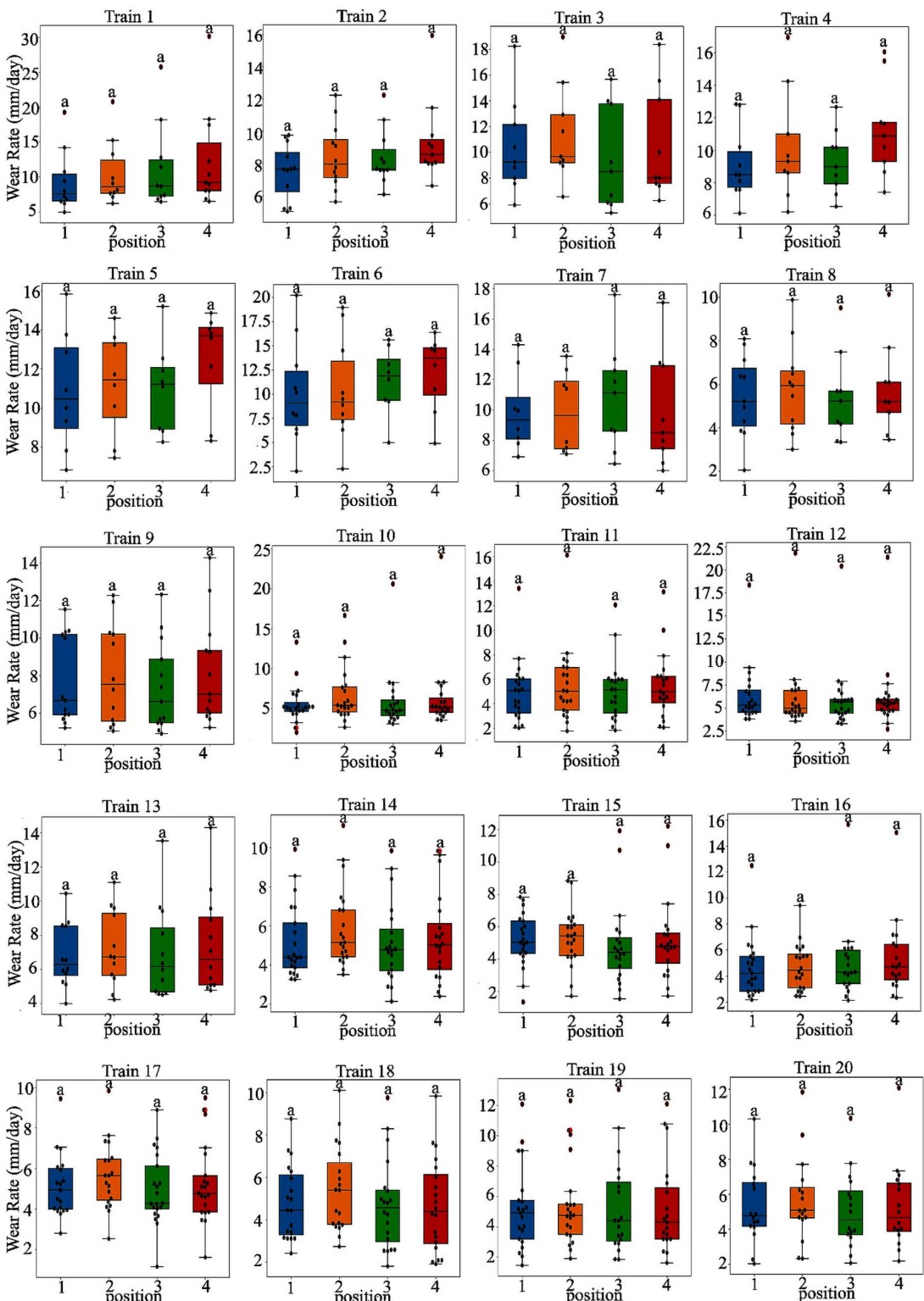

**Fig 7. Box-and-whisker plots showing the mean daily wear rate for strip positions 1–4.** The x-axis denotes the position, and the y-axis represents the wear rate in millimeters per day. Outliers are marked with red dots. Statistical groupings are indicated by Compact Letter Display labels above each box, with matching letters suggesting no significant difference between positions.The study assessed the influence of manufacturers on wear rates, but

faced challenges in making a valid comparison due to the exclusive use of specific manufacturers by certain trains (e.g., Trains 14–17 exclusively used Manufacturer 4). As a result, the manufacturer variable was omitted from the variance analysis, with train identity identified as the key factor driving variations in wear rates.

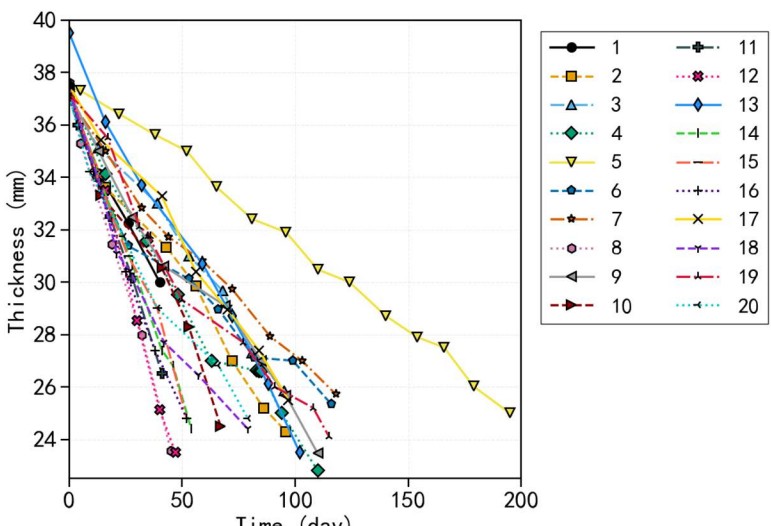

**Fig 8. Degradation of thickness over time for 20 sampled strips.** The x-axis represents the number of days, and the y-axis shows thickness in millimeters. The color-coded lines depict the variability in wear progression rates observed within the fleet.

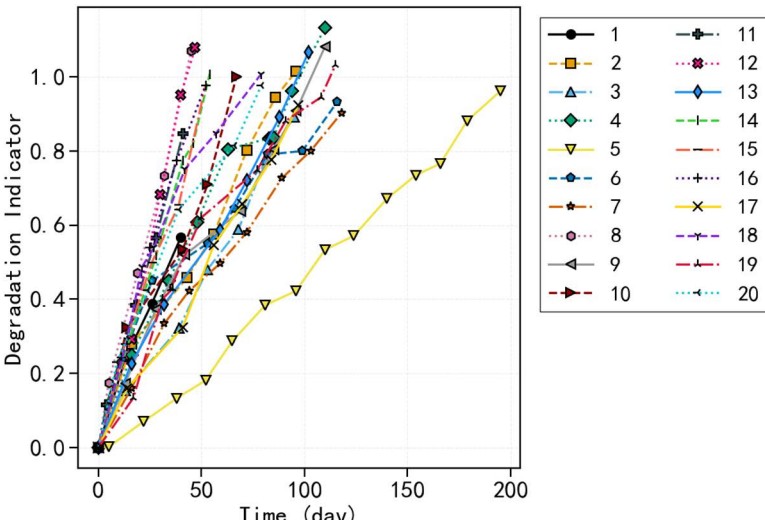

**Fig 9. Degradation indicator trends derived from sliding strips of various trains.** The x-axis represents time elapsed in days, and the y-axis displays a dimensionless degradation indicator ranging from 0 to 1. Each curve corresponds to a specific test sample.

To evaluate the optimal alignment of conventional and random-effect Gamma processes, a hypothesis test is performed under two critical assumptions: (1) the independence of historical degradation paths and (2) the uniform and independent distribution of increments within each path. The methodology for this test is outlined in reference [37].

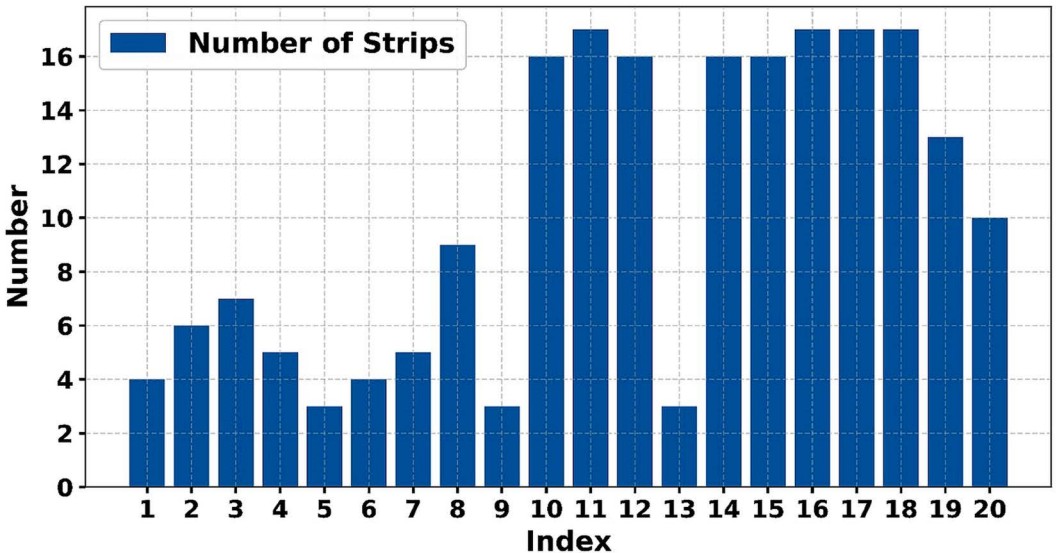

**Fig 10. Number of previous sliding-strip replacements for each pending strip.** The horizontal axis lists the twenty pending samples detailed in Section 3, and the vertical axis shows the count of past strips with the same static attributes. Two primary Gamma-based degradation models commonly used in reliability analysis are the conventional model and the random-effect model [33]. The conventional model assumes that degradation follows a stationary process with consistent behavior across strips. In contrast, the random-effect model allows for variations in degradation rates among different strips. Let $t_{ij}$ represent the measurement time of the j-th observation for the i-th sliding strip. Consistent with previous studies [34], we define the time increment as $\Delta t = t^\gamma_{ij+1} - t^\gamma_{ij}$, where time follows a power-law relationship controlled by a positive parameter γ. The corresponding decrease in the degradation indicator between consecutive observations is denoted as $\Delta y$, which has been empirically shown to display stochastic behavior.

Null Hypothesis (H₀): **β** is unknown but treated as non-random.

Alternative Hypothesis (H₁): **β** is treated as a random variable.

Under the constraints outlined in Equations (2) and (3), γ is assigned a value of 1 for H₀ and 2 for H₁ to simplify the model, reducing the number of independent parameters. This simplification improves computational stability and aids in predicting the RUL. The likelihood functions for each model are described in Equations (4) and (5), where $\Delta y_{ij}$ represents the difference in the degradation indicator between consecutive measurements.

$$L_1(\alpha, \beta) = \prod_{i=1}^{m} \prod_{j=1}^{n_i-1} \frac{\beta^{\alpha(t_{ij+1}-t_{ij})} \Delta y_{ij}^{\alpha(t_{ij+1}-t_{ij})-1} e^{-\beta \Delta y_{ij}}}{\Gamma(\alpha(t_{ij+1}-t_{ij}))} \tag{4}$$

$$L_2(\alpha, \eta, \delta) = \prod_{i=1}^{m} \prod_{j=1}^{n_i-1} \frac{\Delta y_{ij}^{\alpha(t_{ij+1}^2-t_{ij}^2)-1} \delta^\eta \Gamma(t_{ij+1}^2 - t_{ij}^2 + \eta)}{\Gamma(t_{ij+1}^2 - t_{ij}^2)\Gamma(\eta)(\Delta y_{ij} + \delta)^{\alpha(t_{ij+1}^2-t_{ij}^2)+\eta}} \tag{5}$$

and the decision criterion relies on the maximum log-likelihood difference:

$$v = \max \log L_2(\alpha, \eta, \delta) - \max \log L_1(\alpha, \beta) \tag{6}$$

If the statistic $2v$ equals or exceeds a critical value (the 1-θ quantile of a chi-square distribution with 1 degree of freedom [38]; for instance, $2v = 3.8415$ at a .95 significance level), then H₀ is rejected in favor of H₁, and the random-effect model is selected. Otherwise, H₀ is not rejected, and the standard model is retained. Nonlinear optimizations required for maximum

likelihood estimation are conducted using MATLAB's IPOPT solver [39]. The outcomes of the model selection test are outlined in Table 3: solely strips 9, 12, and 19 exhibit a statistically significant value to justify the random-effect model, while the standard model adequately characterizes all remaining strips.

Table 3 presents the results of model selection for sliding strips employing a critical value of 3.8415 at a significance level of .95 in the chi-square distribution. The column "Maximum Log-Likelihood Difference $2v$" shows the computed variances, and the column "Selected Model Index" indicates the favored model (1 = Conventional Gamma, 2 = Random-Effect Gamma).

Depending on the chosen model, we subsequently employ a Bayesian parameter estimation method (as outlined in Cases 1 and 2) to incorporate prior knowledge and new pending data.

### 4.2 Estimation of model parameters via bayesian inference

In Case 1, with Model Index set to 1, the shape parameter α and scale parameter β are assumed to be independent, each following a Gamma prior distribution. The posterior distribution of the parameters and the likelihood function for new observations are represented by equations (7) and (8), respectively.

$$\pi'(\alpha, \beta | Y_i, \bar{t}_i) \propto L_1(Y_i, \bar{t}_i | \alpha, \beta) \pi(\alpha) \pi(\beta) \tag{7}$$

$$L_1(Y_i, \bar{t}_i | \alpha, \beta) = \prod_{j=1}^{[pn_i]-1} \frac{\beta^{\alpha(t^*_{ij+1}-t^*_{ij})} \Delta y_{ij}^{*\alpha(t^*_{ij+1}-t^*_{ij})-1} e^{-\beta \Delta y^*_{ij}}}{\Gamma(\alpha(t^*_{ij+1}-t^*_{ij}))} \tag{8}$$

Let $Y_i$ and $\bar{t}_i$ represent the observed data vector for the pending strip, with asterisks denoting that this data is not included in the historical training set. By incorporating these observations into the likelihood function, we derive an updated expression $L_1$ that is influenced by the new data. In Equation (8), the variable p denotes the percentage of the strip's life cycle completed at the time of prediction. Higher values of p signify a larger amount of in-service data accessible for the strip.

In order to assess the probability, it is necessary to establish prior distributions for α and β. We employ an iterative method to derive informative priors from the historical dataset.

1. Initialization: Initialize **α** and **β** as Gamma-distributed random variables; set iteration counter $i = 1$.

2. Data Extraction: Collect historical samples (degradation paths) with attributes matching the pending strip.

3. Optimization was conducted utilizing the IPOPT solver to maximize the likelihood as defined in Equation (4) for the standard Gamma model by iteratively adjusting the parameters $\alpha_i$ and $\beta_i$. A convergence tolerance was $10^{-6}$, with a maximum of 20 iterations permitted, incrementing $i$ after each optimization iteration.

Table 3. Results of hypothesis testing.

| No. | $2v$ | Selected model index | No. | $2v$ | Selected model index |
|---|---|---|---|---|---|
| **1** | 1.0034 | 1 | **11** | 0.5852 | 1 |
| **2** | 3.4848 | 1 | **12** | 5.2255 | **2** |
| **3** | 2.0375 | 1 | **13** | 1.8193 | 1 |
| **4** | 0.4435 | 1 | **14** | 2.6227 | 1 |
| **5** | 1.1982 | 1 | **15** | 4.0012 | 1 |
| **6** | 3.1052 | 1 | **16** | 0.5322 | 1 |
| **7** | 1.2216 | 1 | **17** | 2.9714 | 1 |
| **8** | 0.8606 | 1 | **18** | 1.7398 | 1 |
| **9** | 6.8239 | **2** | **19** | 4.0653 | **2** |
| **10** | 0.3978 | 1 | **20** | 2.7518 | 1 |

4. Iteration Check: If $i < 20$, return to Data Extraction with updated parameters; otherwise, proceed.

5. Parameter Compilation: Aggregate the optimized values of $\alpha_i$ and $\beta_i$ from each iteration into a sample dataset.

6. Estimate Gamma Distributions: Fit the combined samples of $\alpha_i$ and $\beta_i$ to Gamma distributions to obtain the final prior estimates of α and β for the forthcoming strip.

Case 2 pertains to Model Index = 2. When opting for the random-effect model, two additional parameters (η and δ) are integrated into both the posterior and likelihood formulations to characterize the distribution of the random effect β. Equations (9) and (10) delineate the posterior distribution and likelihood function for this specific case. The estimation procedure in Case 2 mirrors that of Case 1, albeit with the inclusion of an additional parameter, resulting in the estimation of three parameters (α, η, δ) instead of two.

$$\pi'(\alpha, \eta, \delta | Y_i, \bar{t}_i) \propto L_2(Y_i, \bar{t}_i | \alpha, \eta, \delta)\pi(\alpha)\pi(\eta)\pi(\delta) \tag{9}$$

$$L_2(Y_i, \bar{t}_i | \alpha, \eta, \delta) = \prod_{j=1}^{[pn_i]-1} \frac{\Delta y_{ij}^{\alpha(t_{ij+1}^{*2}-t_{ij}^{*2})-1}\delta^\eta \Gamma(\alpha(t_{ij+1}^{*2}-t_{ij}^{*2})+\eta)}{\Gamma(\alpha(t_{ij+1}^{*2}-t_{ij}^{*2}))\Gamma(\eta)(\Delta y_{ij}^* + \delta)^{\alpha(t_{ij+1}^{*2}-t_{ij}^{*2})+\eta}} \tag{10}$$

Fig 11 and Fig 12 displays Gamma prior distributions obtained through Bayesian updating for the model parameters of individual pending strips. Each subplot corresponds to a particular pending strip and presents the estimated parameter distribution based on historical data.

**4.3 Remaining useful life estimation and prediction error analysis**

To dynamically solve Equations (7) and (9), we utilize Markov Chain Monte Carlo (MCMC) techniques through the DREAM toolbox [40,41], enabling effective sampling and convergence evaluation. The toolbox provides three point-estimation approaches for model parameters based on posterior samples [42]:

1. Maximum A Posteriori (MAP): selects parameter values maximizing posterior probability;

2. Maximum Likelihood Estimation with Posterior Constraints (MLE): optimizes the likelihood function under specified posterior constraints;

3. Mean Estimate of Parameters (MEP): uses the mean of the posterior parameter sample.

These methodologies form the basis for assessing estimation techniques in predicting the RUL. After estimating parameters through these approaches, the RUL of the strip is calculated directly. In the conventional model, the RUL at the j-th observation time of the i-th strip is determined by Equation (11) [43,44].

$$R_{ij} = \left[\frac{(1-y_{ij}^*)\beta}{\alpha} + \frac{1}{2\alpha}\right]^{\frac{1}{\gamma}} \tag{11}$$

Let $y_{ij}^*$ represent the degradation indicator at time t. Assuming the scale parameter β follows a probability distribution with parameters η and δ, the mean value $E[\beta] = \eta\delta$ can be utilized to determine the RUL [45]. Thus, the predicted RUL of the component can be calculated as follows:

$$R_{ij} = \left[\frac{(1-y_{ij}^*)\eta\delta}{\alpha} + \frac{1}{2\alpha}\right]^{\frac{1}{\gamma}} \tag{12}$$

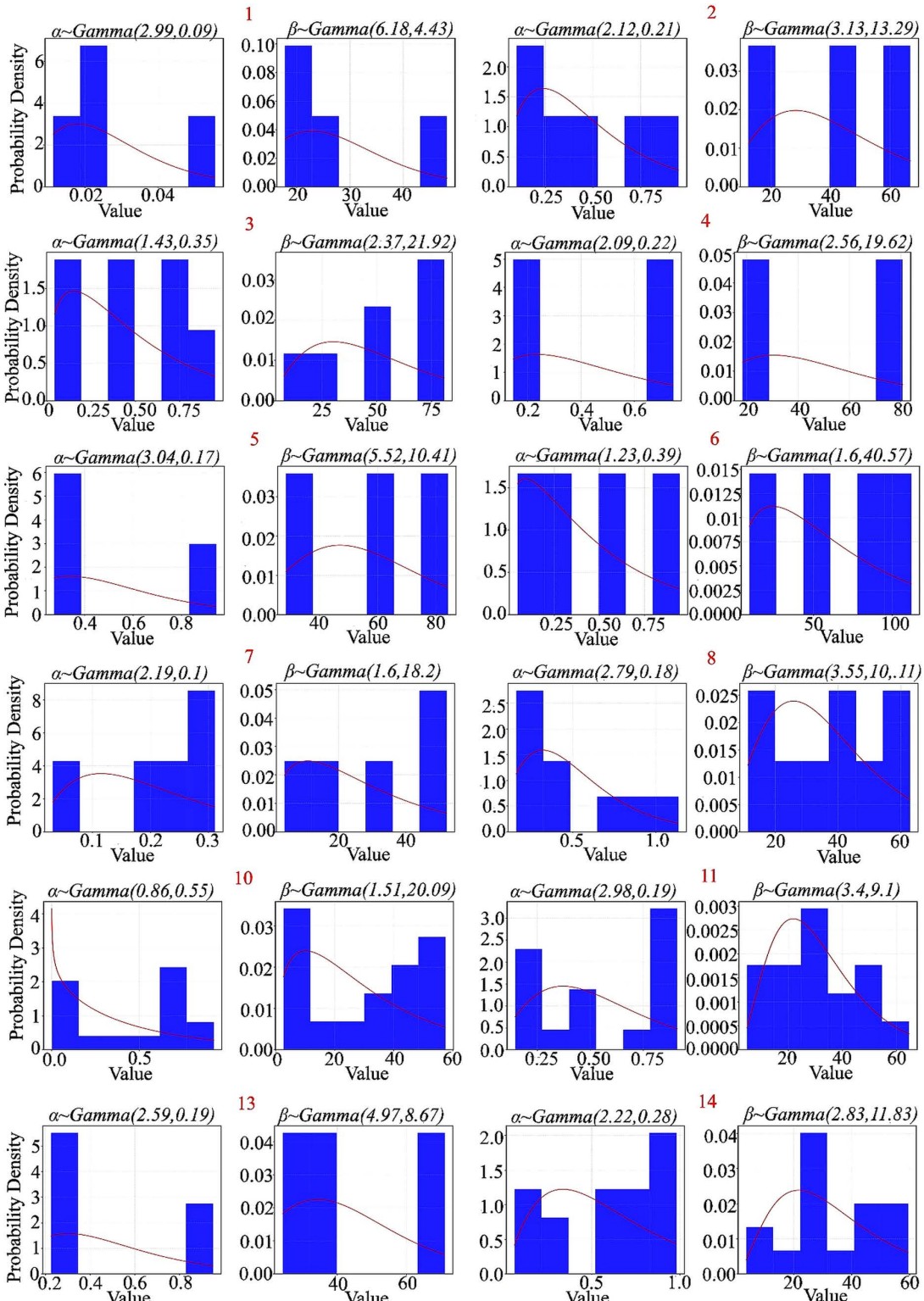

**Fig 11. Gamma prior distributions of model parameters for selected samples.** Each subplot presents a histogram (blue bars) of parameter estimates, including shape (α) and scale (β), obtained from historical data, along with a fitted Gamma probability density function (red curve). These prior distributions serve as the basis for Bayesian updates as additional data is acquired.

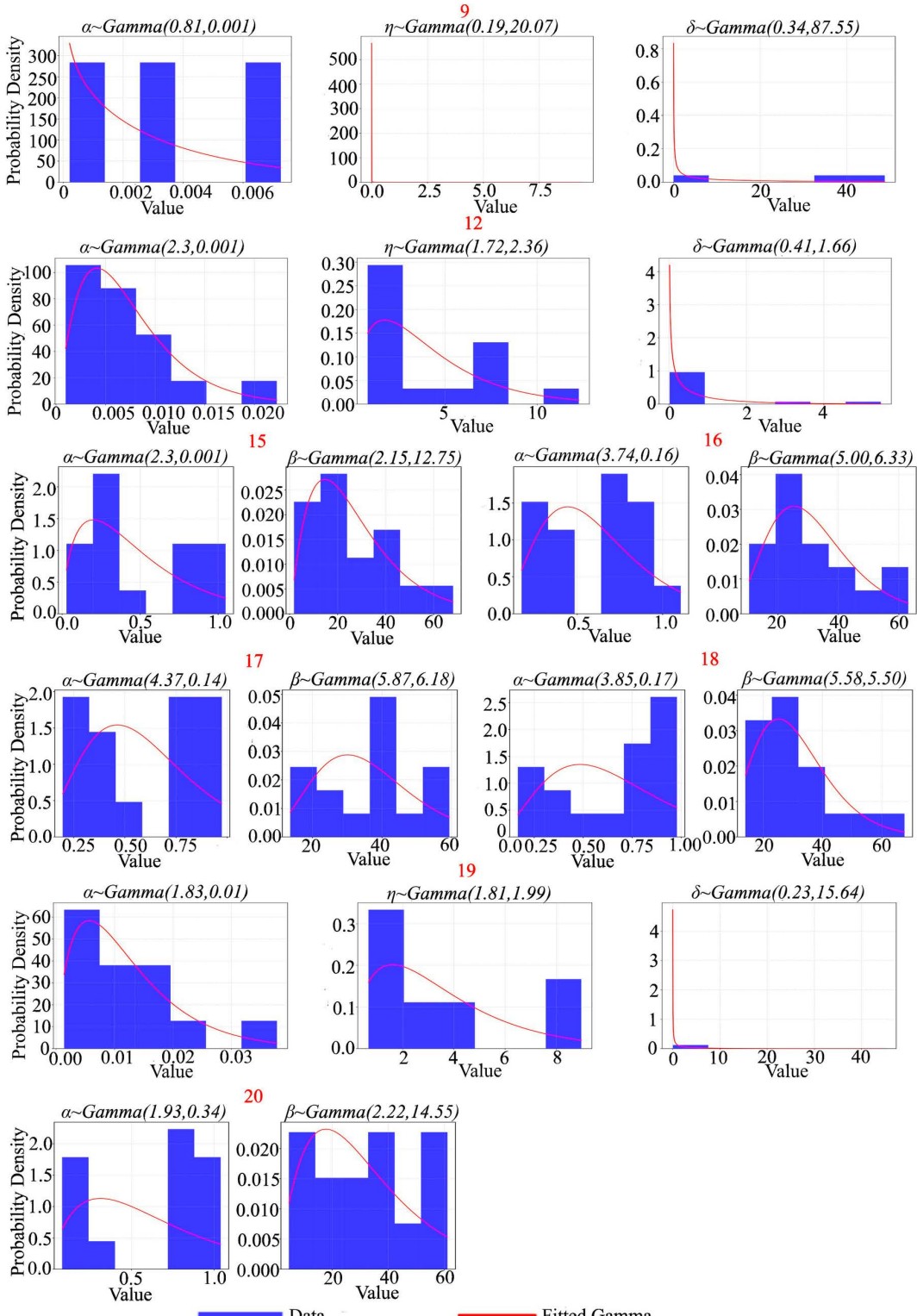

**Fig 12. Gamma prior distributions of model parameters for selected samples.** Each subplot presents a histogram (blue bars) of parameter estimates, including shape (α) and scale (β), obtained from historical data, along with a fitted Gamma probability density function (red curve). These prior distributions serve as the basis for Bayesian updates as additional data is acquired.

If the degradation indicator of a test strip does not reach the specified threshold value of 1 prematurely, denoted as $y_{in_i}^*$, Equations (11) and (12) should be adjusted by replacing the numerical value "1." When considering a fixed parameter undefined in the two model scenarios, the final expressions for RUL are unified into Equation (13).

$$R_{ij} = \begin{cases} \frac{(y_{in_i}^* - y_{ij}^*)\beta}{\alpha} + \frac{1}{2\alpha} & \beta \ is \ not a random variable \\ \left[ \frac{(y_{in_i}^* - y_{ij}^*)\eta\delta}{\alpha} + \frac{1}{2\alpha} \right]^{\frac{1}{2}} & \beta \ is \ a random variable \end{cases} \tag{13}$$

Fig 13 presents a comparative analysis of the RUL predictions generated by three estimation methods (MAP, MLE, and MEP) in relation to the actual remaining life of each test strip at different percentages of life cycle completion $p$. This comparison enables a thorough evaluation of predictive precision as the strip's lifespan advances.

To quantitatively assess prediction accuracy, we employ the mean absolute percentage error (MAPE) as the principal performance measure, as defined in Equation (14).

$$\psi_p = \frac{1}{h} \sum_{i=1}^{h} \frac{\left| R_{i[n_i p]} + t_{i[pn_i]}^* - t_{in_i}^* \right|}{t_{in_i}^*} \tag{14}$$

The variable $\psi_p$ denotes MAPE at a specific percentage of the life cycle, p, while $h$ represents the number of test samples. Fig 14 illustrates the fluctuations in prediction errors (MAPE) for MAP, MLE, and MEP at various p values. As anticipated, the overall error decreases as more monitoring data becomes available, corresponding to higher p values.

## 5 Maintenance scheduling based on remaining useful life predictions

### 5.1 Predictive maintenance criteria and cost-benefit calculation methodology

The adoption of predictive maintenance approaches has empowered metro operators to enhance the scheduling of sliding strip inspections. Traditionally, these inspections were integrated into standard vehicle maintenance procedures. Yet, with the advent of projected RUL data, it is now feasible to proactively plan additional maintenance tasks for the sliding strips beyond regular maintenance cycles. Maintenance choices are systematically modeled during each scheduled vehicle inspection to guarantee efficient resource distribution and enhance system dependability.

The decision to perform extra maintenance on a sliding strip between the j-th and (j + 1)-th scheduled maintenance events depends on a binary decision variable $\lambda_i$, indicating the need for additional maintenance. This decision is determined by Equation (15).

$$\lambda_j = \begin{cases} 0 & R_{ij} \geq t_{ij+1}^* - t_{ij}^*, j = 1, \cdots\cdots, n_i - 1 \\ 1 & R_{ij} < t_{ij+1}^* - t_{ij}^*, j = 1, \cdots\cdots, n_i - 1 \end{cases} \tag{15}$$

Predicting the remaining lifespan of the sliding strip before it reaches the critical wear threshold is essential for optimizing material efficiency and extending its service life. Conversely, once the strip has exceeded this threshold, further maintenance is deemed unnecessary. Notably, these scenarios hinge on a singular maintenance intervention.

The maintenance cost ($V_i^C$) is defined as the total expenditure on additional maintenance tasks, calculated using Equation (16).

$$V_i^C = \begin{cases} \frac{\phi\omega\tau}{\sigma}\left(1 + \sum_{j=1}^{n_i-1} \lambda_j\right) & y_{in_i}^* < 1 \\ \frac{\phi\omega\tau}{\sigma} \sum_{j=1}^{n_i-1} \lambda_j & y_{in_i}^* \geq 1 \end{cases} \tag{16}$$

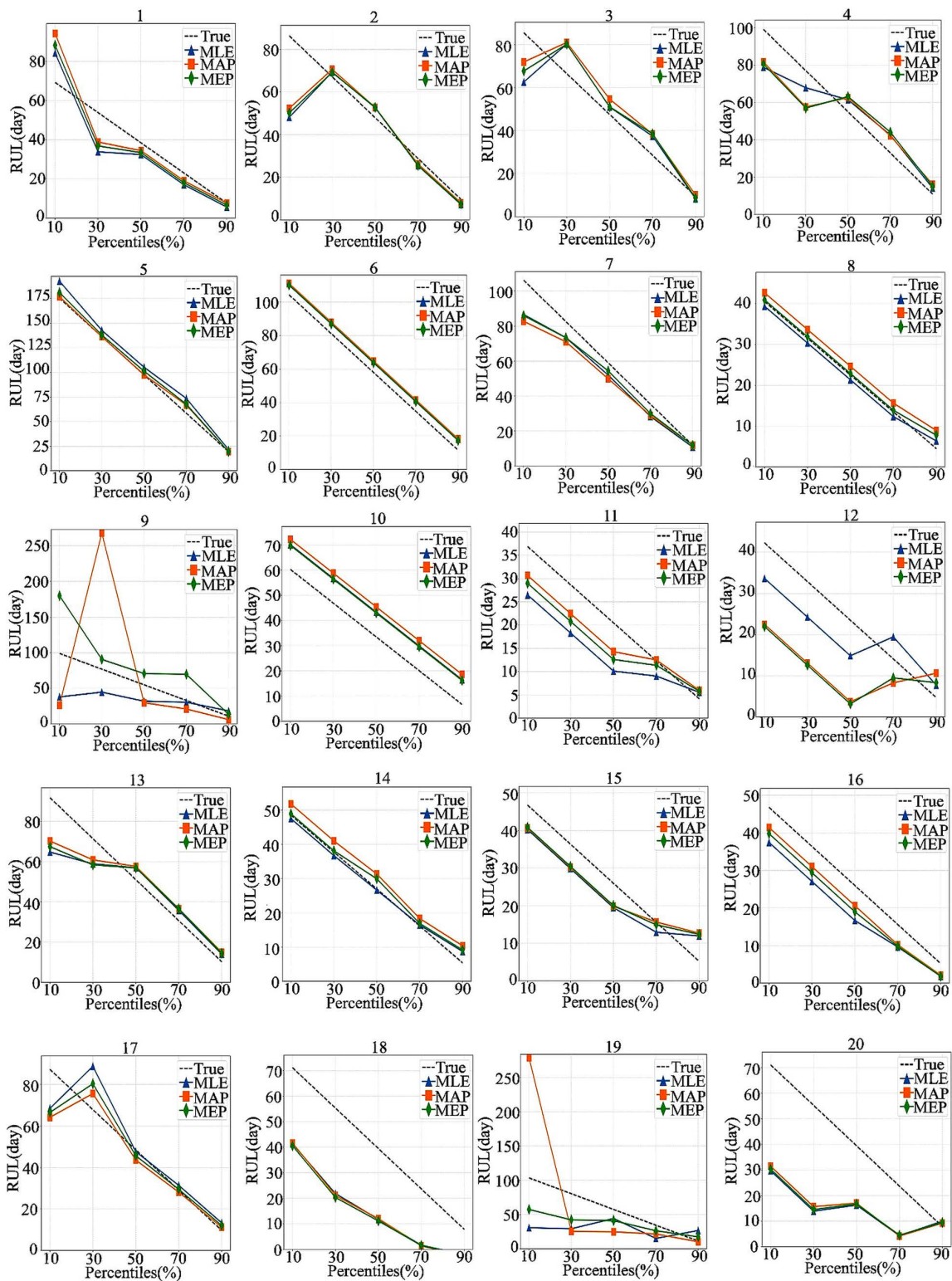

**Fig 13. Comparative performance of predicting RUL at various life cycle completion percentages.** Each subplot displays RUL in days on the vertical axis and the percentage of completed life cycle (p) on the horizontal axis. The lines labeled "True," "MLE," "MAP," and "MEP" represent actual remaining life and RUL estimates derived from three distinct parameter estimation techniques.The three parameter estimation methods exhibit a general

convergence toward the true remaining life as more data are collected over the course of the life cycle, as illustrated in Fig 12. However, during the early-to mid-life stages when data are limited, the prediction errors tend to be larger, with a few exceptions, such as test samples 5, 6, and 8, which deviate slightly from this overall trend. Furthermore, the MAP method demonstrates greater variability compared to both MLE and MEP, suggesting that it is less robust, particularly evident in the higher errors observed for samples 9 and 19 when using the MAP estimates.

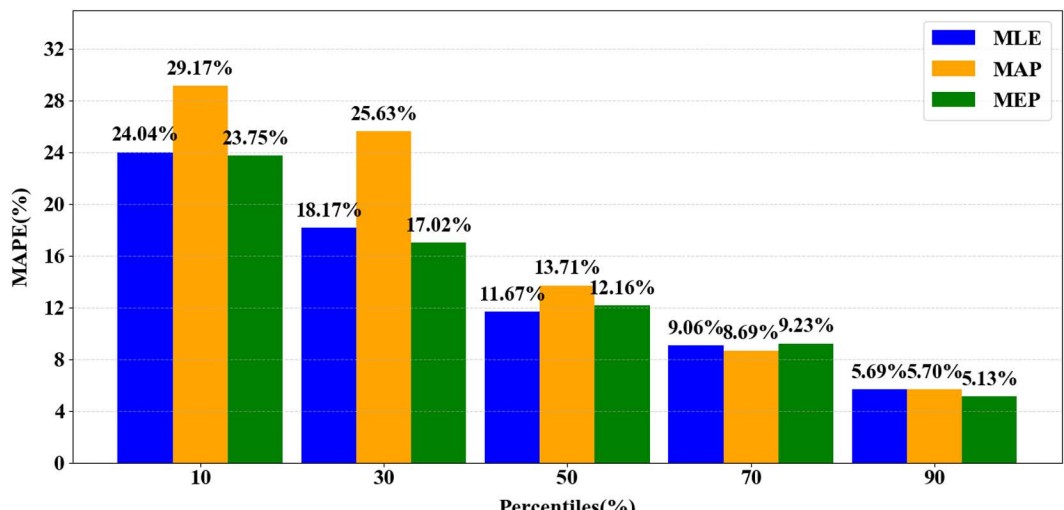

**Fig 14. Correlation between MAPE and life cycle completion percentage (p).** The x-axis represents the life cycle completion percentage, and the y-axis denotes MAPE. The graph shows a decrease in error as the amount of monitoring data increases.These results underscore the substantial influence of the parameter estimation technique on predictive precision in our particular investigation. The MEP consistently yields the most accurate predictions at key stages of the life cycle (10%, 30%, and 90%), indicating its specific suitability for early-life, mid-life, and late-life phases. The MLE method provides the most precise and dependable estimates around the midpoint of the strip's lifespan, typically at 50%. In contrast, the MAP method generally demonstrates higher prediction errors compared to other approaches, except near the 70% threshold, suggesting a lower level of robustness relative to MLE and MEP.

In the equation provided, $\phi$ denotes team size, $\omega$ represents average monthly salary per worker, $\tau$ signifies average maintenance duration in hours, and $\sigma$ stands for monthly working hours per worker. This equation quantifies the labor costs associated with performing extra maintenance activities.

Utilizing RUL prediction can prolong the operational lifespan of a sliding strip by preemptively replacing it before it reaches its wear threshold. This practice enhances material efficiency with a quantifiable probability. The anticipated advantage, labeled as $V_i^S$, is formally articulated in Equation (17).

$$V_i^s = \begin{cases} \frac{(1-\psi_{0.9})R_{in_i}}{E_D}C_1 & y_{in_i}^* < 1 \\ 0 & y_{in_i}^* \geq 1 \end{cases}$$

(17)

Where $C_1$ represents the market price of the sliding strip, $E_D$ denotes the average lifespan of the replaced sliding strips in the historical dataset, and $\psi_{0.9}$ (representing the 90th percentile prediction error) signifies the probability of the predicted RUL deviating from the actual remaining life by an acceptable margin. The choice of the 90th percentile is justified by the prevalent practice of extending a strip's usefulness primarily in its later stages of life, making this threshold a practical choice.

A reduction in strip thickness below a critical threshold prior to scheduled maintenance can result in operational instability. While RUL predictions can mitigate this risk, they are not infallible. Overreliance on forecasts for extending strip usage may result in surpassing the wear threshold. As noted in the Introduction, evaluations of benefits do not consider potential

costs associated with operational disruptions, given the low likelihood of accidents due to delayed replacements. Consequently, beyond the defined threshold, the analysis focuses solely on identifying potential improvements without assigning monetary value to them.

The expected net benefit is determined by subtracting the maintenance cost from the aggregated expected benefits across all samples (denoted as $\sum_{i=1}^{h} V_i^S - V_i^C$).

### 5.2 Parameter specification and analysis of predictive maintenance scheduling results

Accurate assessment of these metrics necessitates precise definition of parameter values. The parameter $\psi_{0.9}$ is derived from our predictive model, reflecting the error estimations outlined in Section 3 (Fig 10). Results for MLE, MAP, and MEP suggest that stands at around 5.69%, 5.70%, and 5.13%, respectively. In contrast, historical data informs the parameter $C_1$, with an unspecified cost for a sliding strip estimated at approximately ¥5,000 based on industry surveys. The study draws upon data from Chongqing Metro Line 6 as a benchmark. Please consult Table 4 for a detailed compilation of parameter values.

The results of planned maintenance according to RUL forecasts utilizing MLE, MAP, and MEP methods are illustrated in Fig 15 and Fig 16. Scheduled maintenance events for vehicles are depicted by black circles, while projected maintenance actions from each parameter estimation technique are shown as colored triangles, with each color corresponding to the specific method as delineated in the legend.

Predicting the remaining useful life when the degradation indicator from the most recent maintenance event reaches or exceeds one is considered unnecessary. Utilizing forecasts from the maintenance event prior to the last one can effectively reduce the likelihood of surpassing the threshold. For example, in Sample 2, where the final degradation indicator exceeds one, all approaches suggest maintenance on day 92, preempting the scheduled maintenance on day 96 and preventing threshold exceedance. Comparable results are evident in Samples 4 and 13.

The MLE, MAP, and MEP methods generally provide consistent predictions for maintenance timing, with occasional deviations. For instance, in Sample 12 (final indicator value of.95), the estimated maintenance timing falls short of preventing threshold exceedance. In Sample 8, both the MLE and MEP methods recommend an additional maintenance step to avoid surpassing the threshold, unlike the MAP method. Conversely, in Sample 9, MAP and MEP suggest an extra maintenance step to prevent threshold exceedance, while MLE does not. Notably, Sample 18, with an actual indicator of.75 on day 42, necessitates maintenance on day 56. In contrast, Samples 14 and 19 exhibit no discrepancies, as all three methods propose similar maintenance schedules. Overall, the three methods exhibit a high level of agreement in anticipated maintenance timing, with only a few exceptions. Further analysis involved calculating costs, benefits, and relevant metrics using Equations (16) and (7), with results detailed in Table 5. Combined, the three approaches require 19 additional maintenance tasks, incurring a labor cost of ¥56.25 each, totaling approximately ¥1,068 in labor expenses. Discrepancies in projected maintenance timing and prediction accuracy result in varying expected benefits, with the MEP

**Table 4. Parameters value and unit.**

| Parameter | Value | Unit |
|---|---|---|
| $\omega$ | 6,000 | ¥ |
| $\phi$ | 3 | person |
| $\tau$ | 0.5 | hour |
| $\sigma$ | 160 | hour |
| $C_1$ | 5,000 | ¥ |
| $E_D$ | 81.28 | day |

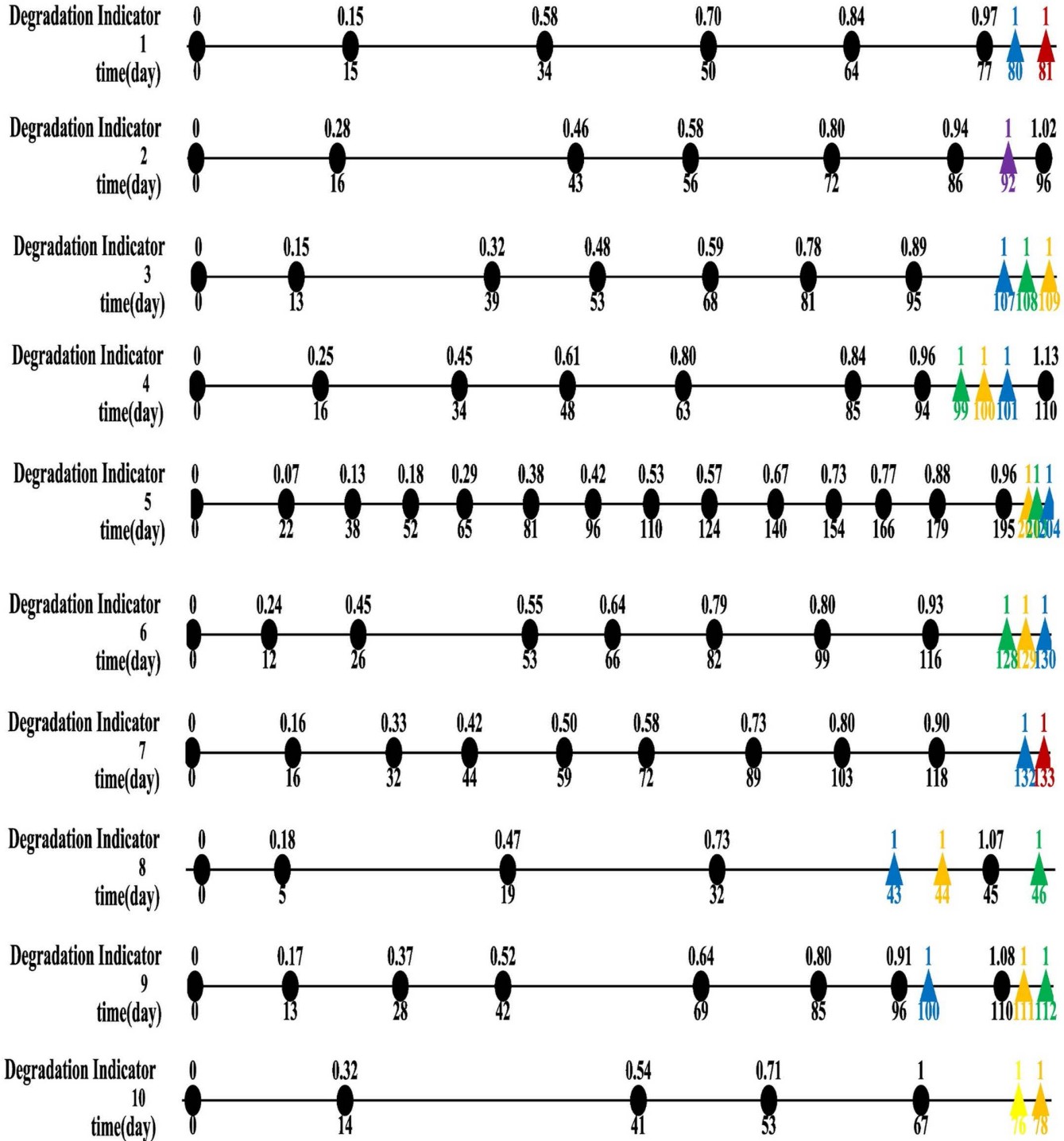

**Fig 15. Scheduled maintenance timelines for top 10 sliding strips based on RUL forecasts.** Each line represents maintenance events marked by black circles, with adjacent numbers indicating the corresponding maintenance dates. Anticipated maintenance occurrences, predicted through various estimation techniques, are represented by colored triangles, with distinct colors indicating temporal intersections as per the legend's specifications.

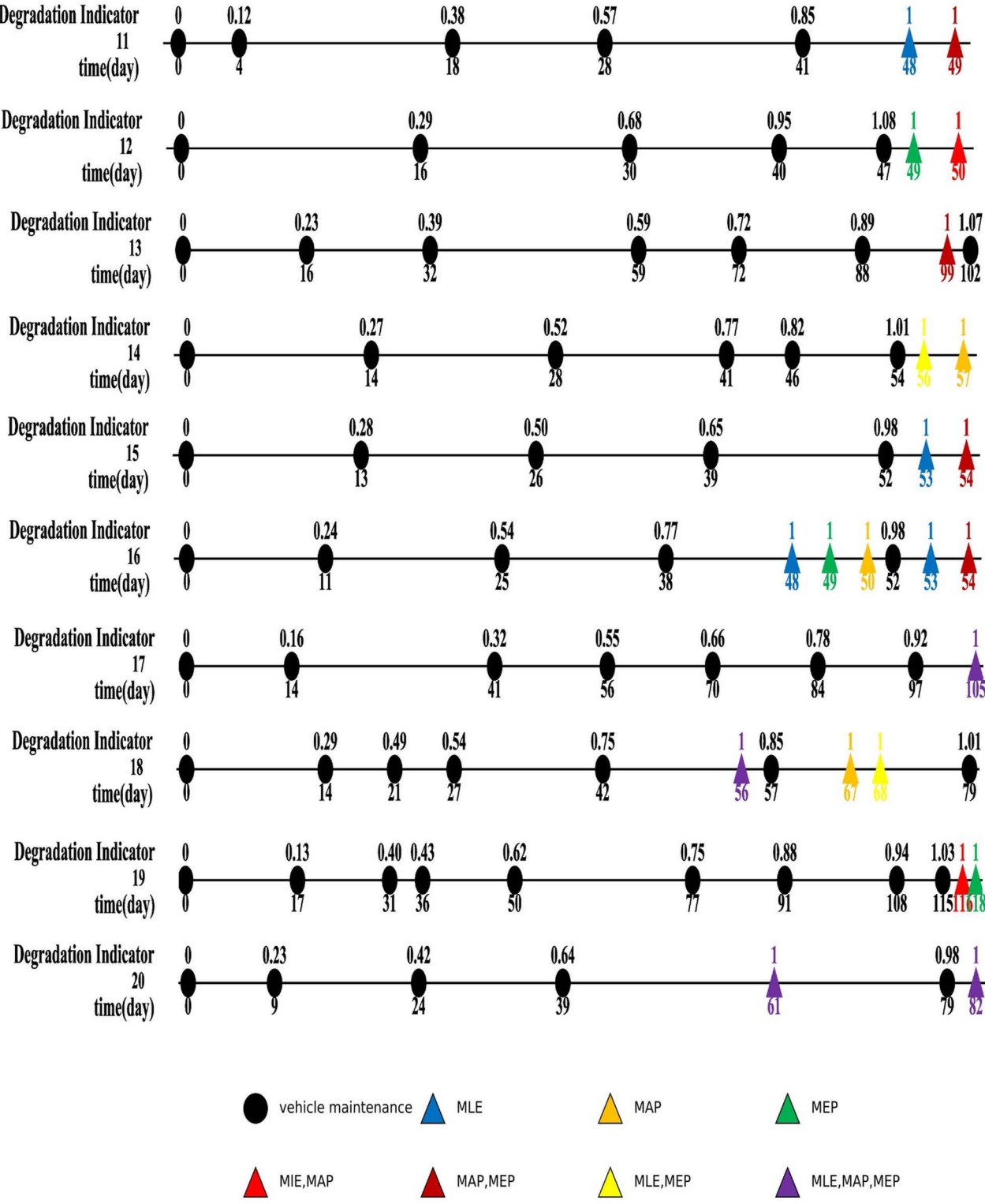

**Fig 16. Scheduled maintenance timelines for last 10 sliding strips based on RUL forecasts.** Each line represents maintenance events marked by black circles, with adjacent numbers indicating the corresponding maintenance dates. Anticipated maintenance occurrences, predicted through various estimation techniques, are represented by colored triangles, with distinct colors indicating temporal intersections as per the legend's specifications.

In instances where the final degradation indicator is below one, it is essential to conduct RUL prediction and determine maintenance actions during the last scheduled maintenance period. Specifically, Samples 1, 3, 5, 6, 7, 11, 15, 16, 17, and 20 are classified in this category. In such cases, besides the potential extension of the service life at the end (which necessitates an additional maintenance task), no extra maintenance is required in preceding intervals. Nevertheless, Samples 16 and 20 present unique situations. For example, Sample 16 reaches a degradation indicator of.77 on day 38, triggering scheduled maintenance on day 48 (MLE), 49 (MEP), or 50 (MAP). Conversely, Sample 20, with a degradation indicator of.64, indicates that maintenance should be executed on day 61 according to all three approaches.

**Table 5. The results of Cost-Benefit and Improvement Metrics.**

| Method | extra Cost(¥) | Anticipated benefit(¥) | Expected net benefit(¥) | Number of Improvements | Number of Non-Improvement |
|--------|---------------|------------------------|-------------------------|------------------------|---------------------------|
| MLE    | 1068          | 4177                   | 3108                    | 6                      | 4                         |
| MAP    | 1068          | 4234                   | 3165                    | 4                      | 6                         |
| MEP    | 1068          | 4117                   | 3048                    | 5                      | 5                         |

method showing the lowest projected net gain. Nevertheless, for decision-making purposes, using the lowest expected benefit as a conservative benchmark is recommended.

In scenarios where the wear threshold is exceeded, MLE exhibits the highest performance, while MAP displays the lowest performance, and MEP falls in an intermediate position. Therefore, when the objective is to optimize measurable net benefits or improve operational dependability upon surpassing the threshold, adopting an MEP-centric approach emerges as a balanced strategy for subsequent parameter estimation.

## 6 Discussion

### 6.1 Impact of prior information on predictive accuracy and implementation details

This study does not directly compare RUL prediction errors with previous research for two main reasons. Firstly, our analysis goes beyond solely reducing errors to optimizing maintenance and replacement schedules using RUL estimates. Secondly, the absence of accessible source code in most prior studies makes replication unattainable within the scope of our research.

The Bayesian updating process adheres to a specific protocol where the likelihood function is updated solely when historical data with corresponding attributes are revised, while new observations from the target directly modify the likelihood. This dual-input approach ensures that the posterior distribution incorporates both past patterns and current observations, with prior parameters significantly influencing predictive precision.

The importance of incorporating prior information is highlighted in the evaluation of prediction accuracy, as demonstrated in Fig 17. Omitting prior information consistently results in increased prediction errors across all stages of the life cycle. Without prior information, MAP and MLE methods produce identical results, as the posterior distribution reduces to the likelihood function alone. This effect is substantial, with prediction errors increasing by 42.07%, 39.31%, 37.27%, 24.85%, and 10.91% at life cycle completion thresholds of 10%, 30%, 50%, 70%, and 90%, respectively. These results unequivocally emphasize the critical importance of incorporating historical knowledge.

### 6.2 Emerging trends in predictive maintenance and cross-domain applicability

Recent technological advancements offer significant opportunities to enhance and broaden our approach to predicting RUL. Digital twin technology allows for the creation of real-time virtual replicas of physical assets, enabling detailed simulation and predictive analysis. In the context of pantograph systems, digital twins can incorporate the Gamma process model to simulate wear progression by adjusting parameters based on operational variables. Additionally, the integration

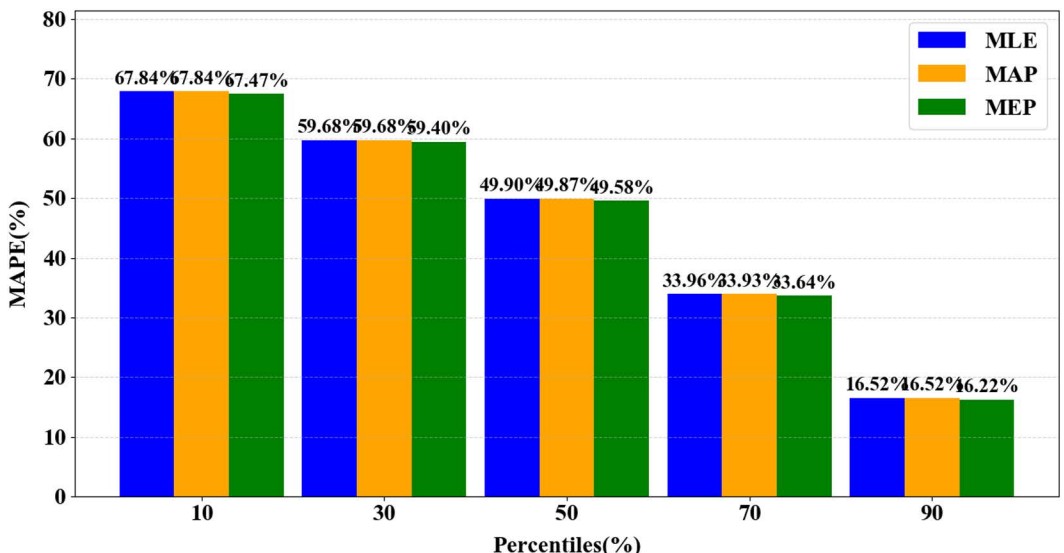

**Fig 17. MAPE at various stages of the life cycle under MLE, MAP, and MEP methods, excluding prior information.** The x-axis represents the life cycle completion percentage, and the y-axis indicates MAPE. The graph shows a reduction in error as additional monitoring data is accumulated, excluding prior information.The framework is operationalized in MATLAB for data processing, employing IPOPT for maximum likelihood estimation and DREAM for Markov Chain Monte Carlo sampling. While alternative programming environments could theoretically support this approach, the adaptation of these specialized optimization and sampling techniques presents practical implementation difficulties. The current implementation requires manual intervention at certain points; nevertheless, all code and data are publicly available on GitHub (https://github.com/jery761210/sliding-strips.git) to foster collaboration and progress.

of diagnostic frameworks that combine data from various sensors, such as vibration signals, acoustic emissions, temperature readings, and electrical data, is essential for comprehensive equipment health evaluations. By amalgamating data from multiple sensor types, the proposed framework can identify degradation patterns that might be missed by single-sensor methodologies.

The methodology under consideration is easily transferable to the examination of analogous transportation systems, like high-speed rail and other transit networks, as evidenced in a recent investigation on gearbox functionality [46]. These systems exhibit fundamental resemblances to pantograph strips, such as uniform wear patterns under different operational loads, standardized inspection procedures, and established safety protocols. Tailoring the model to these alternative systems mainly involves adjusting the data sampling methodology to accommodate distinct operational traits and validate the training data's precision, along with refining pertinent parameters instead of necessitating substantial system restructuring.

## 7 Conclusions

This study presents a framework for predicting the remaining useful life of Metro pantograph sliding strips using a stochastic degradation model grounded in the Gamma process. Analysis of 1,207 sliding strips obtained from Chongqing Metro Line 6 yields four significant findings.

1. Statistical analysis confirms the effectiveness of the Gamma stochastic process in representing the behavior of sliding strip wear, as 97.9% of strips show no significant deviation from the Gamma distribution. This validation underscores the critical importance of choosing the correct degradation model.

2. The predictive accuracy improves steadily as additional condition monitoring data is incorporated. The Mean Estimate of Parameters (MEP) method demonstrates superior performance compared to the Maximum Likelihood Estimation

(MLE) method at 10%, 30%, and 90% life cycle stages, with MLE exhibiting better performance around the 50% milestone. The Maximum A Posteriori (MAP) method generally results in higher errors, except in the proximity of the 70% life cycle stage.

3. Incorporating prior distributions improves prediction accuracy by 10% to 42% across different life stages compared to methods that do not utilize historical data. This substantial improvement highlights the crucial role of leveraging previous knowledge in achieving reliable predictions of Remaining Useful Life.

4. Implementing maintenance scheduling based on Remaining Useful Life (RUL) results in a net gain of approximately ¥3,048 per test case through optimized replacement timing. This financial validation highlights the practical importance of predictive maintenance approaches within the Metro operations framework.

The Gamma process framework harmonizes theoretical robustness with operational utility, enabling precise degradation characterization and enhancing maintenance planning. Through the integration of Bayesian inference with practical maintenance constraints, a methodology is formulated that is readily applicable in practical settings.

Future research should prioritize enhancing the framework's capabilities and validating its effectiveness across various wear mechanisms and operational cycles, such as high-speed rail gearboxes and other transportation assets. The integration of continuous Internet of Things sensor data could facilitate real-time updates on Remaining Useful Life through efficient online Bayesian algorithms. Hybrid models that merge Gamma processes with deep learning could adeptly capture nonlinear degradation patterns while maintaining interpretability. Enhancing computational efficiency via advanced sampling methods or parallel processing would support real-time fleet-wide implementation. By optimizing maintenance activities for critical components, overall lifecycle expenses could be reduced. These advancements have the potential to elevate the prototype into a comprehensive predictive maintenance platform suitable for deployment in urban and high-speed rail networks.

## Acknowledgments

The authors would like to acknowledge the Chongqing Rail Transit Group for their data supports.

## Author contributions

**Conceptualization:** Jie Liu.

**Data curation:** Jie Liu, Chuang Wu.

**Methodology:** Chuang Wu.

**Resources:** Chuang Wu.

**Software:** Jie Liu.

**Writing – original draft:** Jie Liu.

**Writing – review & editing:** Jie Liu.

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
