## [Decision Letter · Decision Letter 0]

Dear Dr. Liu,

Thank you for submitting your manuscript to PLOS ONE. After careful consideration, we feel that it has merit but does not fully meet PLOS ONE’s publication criteria as it currently stands. Therefore, we invite you to submit a revised version of the manuscript that addresses the points raised during the review process.

AE: The reviewers acknowledge the main contributions of this research work. However, some reviewers have raised concerns regarding novelty and technical issues that should be addressed.

We look forward to receiving your revised manuscript.

Kind regards,

Ke Feng

Academic Editor

PLOS ONE

Journal Requirements:

“This work was supported by the Science Technology Research Program of Chongqing Municipal Education Commission (Grant No. KJQN202203405, KJQN202303402, KJQN202203417).”

“The authors would like to acknowledge the Chongqing Rail Transit Group for their data supports.And this work is supported by the Science Technology Research Program of Chongqing Municipal Education Commission (Grant No. KJQN202203405, KJQN202303402, KJQN202203417).”

“This work was supported by the Science Technology Research Program of Chongqing Municipal Education Commission (Grant No. KJQN202203405, KJQN202303402, KJQN202203417).”

5. Please ensure that you refer to Figure 15 in your text as, if accepted, production will need this reference to link the reader to the figure.

**Additional Editor Comments:**

AE: The reviewers acknowledge the main contributions of this research work. However, some reviewers have raised concerns regarding novelty and technical issues that should be addressed.

Reviewers' comments:

Reviewer's Responses to Questions

**Comments to the Author**

1. Is the manuscript technically sound, and do the data support the conclusions?

Reviewer #1: Yes

Reviewer #2: Yes

2. Has the statistical analysis been performed appropriately and rigorously?

Reviewer #1: Yes

Reviewer #2: Yes

3. Have the authors made all data underlying the findings in their manuscript fully available?

Reviewer #1: Yes

Reviewer #2: Yes

4. Is the manuscript presented in an intelligible fashion and written in standard English?

Reviewer #1: No

Reviewer #2: Yes

Reviewer #1: This paper proposes a Gamma process-based framework for predicting the remaining useful life (RUL) of subway pantograph sliding strips using real-world data from Chongqing Subway Line 6. By incorporating both historical and real-time observations through Bayesian inference, the model achieves accurate, stage-aware RUL predictions, with posterior estimation methods compared for robustness. The predicted RUL is further integrated into maintenance scheduling, enabling cost-effective and timely decision-making for subway system upkeep. This paper presents an interesting approach; however, several critical issues remain unresolved:

1. The overall formatting of the manuscript is poor, and the English expression is awkward and unnatural in places.

2. This paper is an engineering-focused work, but the literature review is inadequate. It should focus more on studies related to Remaining Useful Life prediction for the Subway Pantograph Sliding Strips, providing a more relevant context.

3. Real-world data is crucial for RUL prediction tasks and is highly valuable. Therefore, a more detailed description of the dataset's characteristics is necessary to strengthen the paper's foundation.

4. The decision to use the Gamma process instead of the Wiener process is not well justified. The Wiener process is also widely used in RUL prediction tasks, as seen in Reliability Engineering & System Safety (DOI: 10.1016/j.ress.2024.110014). A discussion on why the Gamma process was chosen over the Wiener process should be included.

5. The applicability of the proposed method is unclear. For example, similar operational and maintenance requirements exist in high-speed rail systems (see Vehicle System Dynamics, DOI: 10.1080/00423114.2024.2369718). It would be beneficial to explore whether the proposed method could be applied to such systems.

6. The resolution of all images in the manuscript should be improved to ensure clarity and readability.

Reviewer #2: This paper proposes an adaptive, data-driven framework for predicting the remaining useful life (RUL) of subway pantograph sliding strips using data from Chongqing Subway Line 6. The paper is well-structured and presents promising results. To further enhance the manuscript, the following suggestions are offered:

1. The Abstract could benefit from a clearer emphasis on the practical significance of this research. Highlighting the engineering context and potential real-world applications would help readers better understand the value of the proposed method.

2. While the authors have provided a thorough review of current research, it would be beneficial to more explicitly summarize the existing research gaps before introducing the contributions and novelty of this work. Additionally, expanding the discussion to include emerging trends in machine learning and signal processing—particularly in industrial applications—could strengthen the manuscript. For example, exploring connections to recent advancements in areas such as digital twin methodology for vibration-based monitoring and prediction of gear wear, review of vibration-based gear wear monitoring and prediction techniques, and neuro-fuzzy system-guided cross-modal zero-sample diagnostic framework using multi-source heterogeneous non-contact sensing data could provide valuable context and demonstrate the broader relevance of this research.

3. The resolution of the figures in the manuscript could be improved to ensure clarity and enhance the overall presentation of the results.

4. There are occasional grammatical errors throughout the manuscript. A thorough proofreading to address these issues would improve the readability and professionalism of the paper.

5. Including a section on potential future research directions at the end of the Conclusion would provide a forward-looking perspective and inspire further exploration in this field.

**Do you want your identity to be public for this peer review?** For information about this choice, including consent withdrawal, please see our Privacy Policy

Reviewer #1: No

Reviewer #2: No

---

## [Author Response · Author response to Decision Letter 1]

3 Jun 2025

Because this is a major revision, almost every paragraph in the “Revised Manuscript with Track Changes.docx” has undergone substantial grammatical and stylistic changes, so in most places entire paragraphs are highlighted.

I have included this amended Role of Funder statement in my cover letter.

---

## [Decision Letter · Decision Letter 1]

Predicting the Remaining Useful Life of Metro Pantograph Sliding Strips Using Gamma Processes and Its Implications for Maintenance Scheduling

PONE-D-25-12961R1

Dear Dr. Liu,

We’re pleased to inform you that your manuscript has been judged scientifically suitable for publication and will be formally accepted for publication once it meets all outstanding technical requirements.

Kind regards,

Ke Feng

Academic Editor

PLOS ONE

Additional Editor Comments (optional):

This paper has been improved by addressing the comments from reviewers. The quality of this paper has been improved significantly. The reviewers agree to accept this paper.

Reviewers' comments:

Reviewer's Responses to Questions

**Comments to the Author**

Reviewer #1: (No Response)

Reviewer #2: All comments have been addressed

2. Is the manuscript technically sound, and do the data support the conclusions?

Reviewer #1: (No Response)

Reviewer #2: Yes

3. Has the statistical analysis been performed appropriately and rigorously?

Reviewer #1: (No Response)

Reviewer #2: Yes

4. Have the authors made all data underlying the findings in their manuscript fully available?

Reviewer #1: (No Response)

Reviewer #2: Yes

5. Is the manuscript presented in an intelligible fashion and written in standard English?

Reviewer #1: (No Response)

Reviewer #2: Yes

Reviewer #1: The manuscript has been significantly improved through the revisions. The key issues raised in earlier rounds have been addressed adequately, and the current version demonstrates a clear structure, enhanced clarity, and stronger experimental support. From an academic standpoint, the paper is now suitable for consideration for acceptance.

Reviewer #2: This paper has been revised based on the comments from reviewers. The quality of this paper has been improved significantly. It can be accepted now.

**Do you want your identity to be public for this peer review?** For information about this choice, including consent withdrawal, please see our Privacy Policy

Reviewer #1: No

Reviewer #2: No

---

## [Editor Report · Acceptance letter]

PONE-D-25-12961R1

PLOS ONE

Dear Dr. Liu,

I'm pleased to inform you that your manuscript has been deemed suitable for publication in PLOS ONE. Congratulations! Your manuscript is now being handed over to our production team.

Kind regards,

on behalf of

Professor Ke Feng

Academic Editor

PLOS ONE